# Study on Water Saving Potential and Net Profit of *Zea mays* L.: The Role of Surface Mulching with Micro-Spray Irrigation

**Zhaoquan He [1,2,*], Xue Shang [1] and Tonghui Zhang [2,3]**

[1]  School of Life Sciences, Yan'an University, Yan'an 716000, China; scher1@163.com

[2]  Naiman Desertification Research Station, Northwest Institute of Eco-Environment and Resources, Chinese Academy of Sciences, Lanzhou 730000, China; zhangth@lzb.ac.cn

[3]  Northwest Institute of Eco-Environment and Resources, Chinese Academy of Sciences, Lanzhou 730000, China

\*  Correspondence: hzq@yau.edu.cn; Tel.: +86-150-9540-3883

**Abstract:** Water shortage threatens agricultural sustainability in Horqin Sandy Land, northeast China. To explore the effects of various surface mulching patterns with micro-spray irrigation on the yield, water consumption ($ET_c$), and water-saving potential of maize (*Zea mays* L.), we used three treatments: straw mulching (JG), organic fertilizer mulching (NF), and no mulching (WG; control). In each treatment, plants were supplied with 500 mm of total water (irrigation plus precipitation) during the entire growing season and were irrigated with the amount of total water supply minus precipitation. Yield and water use efficiency (WUE) showed a significant negative correlation with water saving potential per unit yield ($P_y$) and water saving potential per unit area ($S_p$), which were also consistent with their relationships in the function model. Meanwhile, a remarkably positive correlation occurred between yield, WUE, and net economic profit, respectively. The JG treatment, which was mainly affected by light and temperature production potential ($Y_c$), grain yield, and $ET_c$, showed the lowest $P_y$ (0.16 m$^3$ kg$^{-1}$) and $S_p$ (2572.31 m$^3$ hm$^{-2}$), and the maximum increase in yield, WUE, and net economic profit, extending to 16,178.40 kg hm$^{-2}$, 3.25 kg m$^{-3}$, 17,610.09 yuan hm$^{-2}$, respectively, which were significantly higher than those in NF and WG, ($p$ <0.05). Thus, straw mulching with micro-spray irrigation was the best treatment for maximizing yield and WUE. Organic manure mulching and no mulching need further investigation, as these showed high $P_y$ and $S_p$, which were together responsible for lower WUE.

**Keywords:** water resources; surface mulching; water saving potential; micro-spray irrigation; economic profit

---

## 1. Introduction

Maize (*Zea mays* L.) is the second most important grain crop in China. Inner Mongolia is one of the major maize production regions of China that supports the livelihood of farmers in the region [1]. Maize requires a substantial amount of water during the growth phase but 80% of Inner Mongolia is arid and semi-arid, and water shortage in these areas continues to worsen with global climate change [2]. In addition, increase in the cultivable area is remarkably slow; therefore, the crop production area is unlikely to expand in the region so increasing yield per unit area is the only solution to continually support maize-production dependent households [3]. The improvement of maize yield per unit area primarily depends on the extension and reform of agricultural water saving planting technologies [4]. On the one hand, surface mulching is commonly used because of its notable positive impact on water conservation and yield. There are reports that mulching conserves

soil moisture by reducing evaporation and saving 12–84% of the irrigation water [5]. Common surface mulching methods include plastic mulching, straw mulching, organic manure mulching, and biochar mulching [6–8]. Effective use of crop straw and animal manure is conducive to the intensive management of agricultural resources, which is of practical significance for improving crop cropping systems, developing sustainable biodiversity of agro-ecosystems, and implementing national poverty reduction policies [9]. In addition, crop straw mulching and fully rotten organic manure mulching can improve the topsoil physics and chemistry nature of soil remarkably, without causing environmental pollution [10]. On the other hand, efficient irrigation contributes to increases in yields of crops and in income for the local farmers, providing evidence of the significance of irrigation in the past and for future poverty alleviation in China [11]. Therefore, combining mulching with advance irrigation method (drip/micro irrigation) increase more significantly crop productivity and water use efficiency (WUE), reduce water consumption ($ET_c$) in a region where water shortage is the major factor limiting agricultural sustainability [12–14].

The potential and configuration of climate resources (light, heat, and water) affect their utilization and ultimately limit the sustainable development of agricultural production [15]. Therefore, investigation of the characteristics of crop climate production potential is essential. The production potential of light and temperature is the main focus of studies in China. Water saving potential refers to the amount of irrigation water that can be saved per unit scale of crops, involving four scales, that is, crop, field, irrigation area, and regional/basin, respectively, and is a major factor affecting agricultural structure [16,17]. The definition of agricultural water saving potential is mainly based on water efficient methods, various water saving techniques, and management [16,18]. Previously, many approaches have been adopted to measure water saving potential, including efficient irrigation technology and method, irrigation scheduling improvement [19]. For example, some studies used the water balance method and remote sensing technology to calculate water consumption for obtaining the theoretical water saving potential [20,21], while many studies utilized crop models, such as DASSAT and WOFOST [22], to simulate the water consumption of crops and water saving ability, generating favorable results. Additionally, economic profit evaluation of agricultural technology has been largely explored in maize, confirming that the water saving irrigation method is significantly better than traditional irrigation technology [23], and no-till and permanently fixed ridge is better than conventional tillage [24].

Although responses of yield and water consumption of maize to surface mulching have been explored extensively, as mentioned above, research on the water saving potential per unit yield ($P_y$) and water saving potential per unit area ($S_p$) of maize (two angles of water-saving potential calculation, that is, crop scale and area scale, respectively) with surface mulching and micro-spray irrigation technology, and on the primary factors affecting the water saving potential of maize, is lacking. In this study, we used the meteorological observation data of the Naiman desertification station to achieve the following two main objectives: (1) to adjust the crop planting structure and farmland irrigation system through the improvement of the efficiency of irrigated agriculture by determining the surface mulching patterns with the greatest water-saving ability; and (2) to establish the optimal crop planting pattern with high net economic benefits and prominent water saving ability to provide theoretical support for the evaluation measures of economic benefits of water-saving planting patterns of farmland crops, thus expanding the appropriate ecological planting scale in this region.

## 2. Materials and Methods

### 2.1. Experimental Site

The experiment was carried out at the Naiman Desertification Research Station of the Chinese Academy of Sciences (42°58′ N, 120°43′ E; 360 m a.s.l.) in April–September 2017, located in the eastern part of Inner Mongolia Autonomous Region, China (Figure 1). Naiman, located in the southwest of Horqin Sandy Land [25], has a semiarid continental monsoon climate. The experimental site,

with sandy soil texture sensitive to wind erosion, has a mean annual precipitation of approximately 360 mm (265 mm in this growing season), an annual mean evaporation of around 1950 mm, and an annual mean temperature of 6.40 °C, of which the minimum monthly average temperature of −13.50 °C occurred in January and the maximum of 23.80 °C in July. In the soil depth of 0–60 cm, soil organic carbon content, pH (1:2.5 water), and electrical conductivity (1:5 water) at 0–30 cm depth before planting are 2.48 g kg$^{-1}$, 9.23, 62.73 μS cm$^{-1}$, respectively. Field capacity was 12.77%, wilting point was 5.40%, water saturation was 30.24%, saturated hydraulic conductivity was 0.93 mm min$^{-1}$, as well as bulk density being 1.55 g cm$^{-3}$.

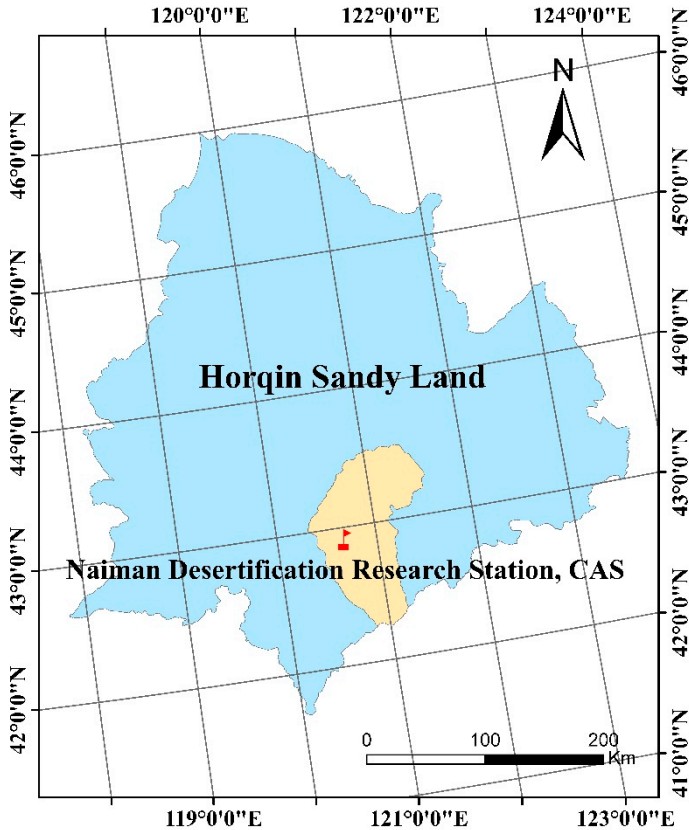

**Figure 1.** Location of Naiman desertification research station in China. Notes: the blue region represents Horqin Sandy Land; the yellow region represents the Naiman banner; the red flag represents Naiman desertification research station, CAS.

*2.2. Test Design*

The trial was laid out in a completely randomized plot design. A total of 51 plots (2 m × 2 m), representing 3 treatments and 17 replicates, were created. The *Zea mays.* L. cv. Jingke 958 variety was chosen as the tested cultivar for all the treatments and was planted by a tube-shaped seeder on 27 April at a depth of 5 cm. The hole of the seeds was 2.5 cm in the direction of the row and was 3.0 cm in the direction of the column, which was reserved for the space of maize growth. The planting density of maize was 60,000 plants hm$^{-2}$. Each plot contained 4 rows, with 6 maize plants per row, that is, 4 rows of 6 columns, at a plant-to-plant spacing of 20 cm and row-to-row spacing of 36 cm (Figure 2). The spacing of 0.5 m between each plot was provided for maintaining independence among treatments, and a buffer channel of 1 m width was provided in the neighborhood of experimental fields to avoid edge effects. The experimental field was oriented west to east. The field was rectangular in shape, with 9 m in the north-south direction and 44 m in the east-west direction. Three treatments were established: straw mulching (JG), organic manure mulching (NF) and no mulching (WG; control).

In the JG treatment, when the maize seedlings reached a height of 20 cm, crushed maize straw was evenly applied to 100% of the soil surface with 4000 kg hm$^{-2}$ (1.6 kg per plot). In the NF treatment, cattle and poultry manure were the sources of organic manure. They were collected from the cattle and poultry farms located in Naiman banner and were uniformly mixed with soil 15 days before applying, making it fully rotten, preventing environmental pollution. Because organic manure that is not fully rotten contains various bacteria, this would cause a high incidence of crop diseases and insect pests, and affect the ecological environment when applied directly to farmland. Physical and chemical properties of the organic manure were, 145.77 g kg$^{-1}$ of soil organic carbon, 260.49 g kg$^{-1}$ of soil organic matter, 10.25 g kg$^{-1}$ of total nitrogen, 8.64 g kg$^{-1}$ of total phosphorous, and 11.57 g kg$^{-1}$ of total potassium. When the maize seedlings reached a height of 20 cm, organic manure was evenly distributed on 100% of the soil surface with 30,000 kg hm$^{-2}$ (12 kg per plot). In the WG (control) treatment, no surface mulching was used during the entire growth period of the maize.

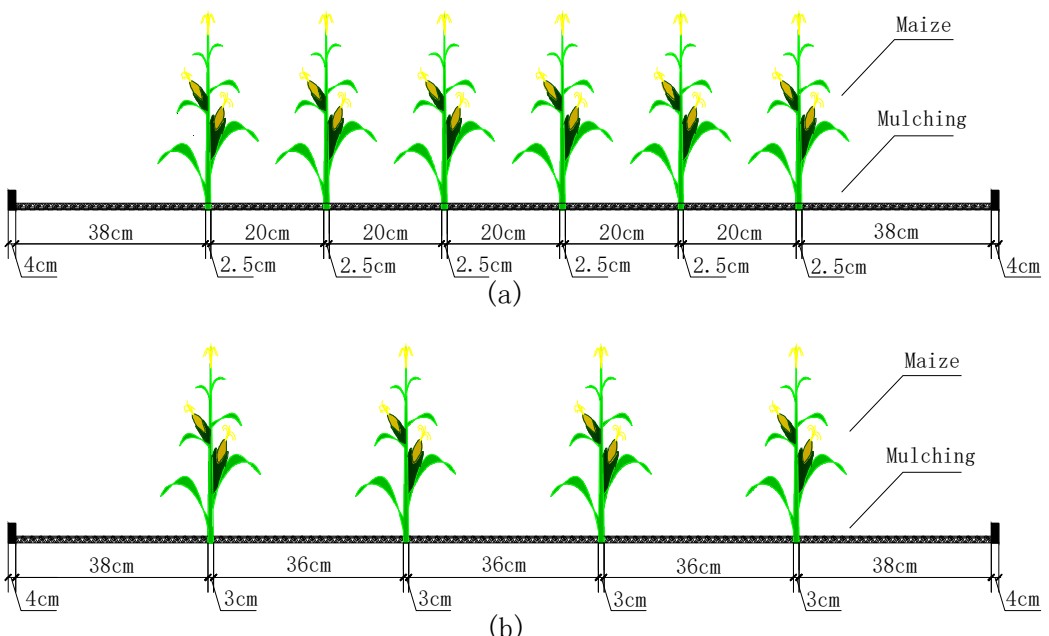

**Figure 2.** Sketch of experimental site. (**a**) represents the front view; (**b**) represents the lateral view.

*2.3. Irrigation Scheme*

In each treatment, zonal micro-spray irrigation was used. Maize plants were supplied with 500 mm of water, including irrigation and precipitation, during the entire growth season. The irrigation amount was the total water supply minus precipitation. The maize growth season was divided into five stages: seeding, jointing, heading, filling, and ripening. According to the water requirements of maize at each growth stage, 15%, 35%, 22%, and 28% of the total water supply was applied to the seedling-jointing, jointing-heading, heading-filling, and filling-ripening stage, respectively [26]. The irrigation level was the same across all treatments. In each growth phase, if the precipitation exceeded the upper limit of the designed water supply, the water increment needed to be subtracted from the next irrigation, ensuring the same total water supply for all treatments throughout the entire growing season. Maize was irrigated on days with no or low wind (<1.5 m s$^{-1}$) to achieve uniform irrigation. Irrigation regimes are summarized in Table 1. Spray lines (42 m) were installed in the middle of the plot along the east–west direction, with a nozzle spacing of 50 cm and discharge rate of 1 L h$^{-1}$. Irrigation groundwater was measured continuously using flowmeters.

**Table 1.** The irrigation regimes across the growing season (mm, April–September 2017).

| Growth Stage | Precipitation | Irrigation Only | Straw Mulching (JG) | Organic Manure Mulching (NF) | No Mulching (WG) |
|---|---|---|---|---|---|
| | 26 April–28 April | | Covers all three treatments: 0.00 | | |
| | | 28 April–1 May | 38.97 | 38.76 | 39.13 |
| Seeding-Jointing stage | 2 May–22 May | | Covers all three treatments: 29.90 | | |
| | | 23 May–25 May | 5.94 | 6.02 | 5.98 |
| | 23 May–8 June | | Covers all three treatments: 0.60 | | |
| | Total Irrigation from Seeding to Jointing Stage | | 74.81 | 74.72 | 75.01 |
| | Irrigation Deviation | | −0.19 | −0.28 | +0.01 |
| | | 9 June–13 June | 87.06 | 86.95 | 87.10 |
| | 14 June–24 June | | Covers all three treatments: 5.50 | | |
| Jointing-Heading stage | | 25 June–27 June | 38.24 | 38.09 | 38.12 |
| | 25 June–15 July | | Covers all three treatments: 54.70 | | |
| | Total Irrigation from Jointing to Heading Stage | | 185.50 | 185.24 | 185.42 |
| | Irrigation Deviation | | +10.50 | +10.24 | +10.24 |
| | | 16 July–20 July | 62.88 | 62.90 | 62.84 |
| Heading-Filling stage | 20 July–28 July | | Covers all three treatments: 35.70 | | |
| | Total Irrigation from Heading to Filling Stage | | 98.95 | 98.60 | 98.54 |
| | Irrigation Deviation | | −11.42 | −11.40 | −11.46 |
| | 29 July–4 August | | Covers all three treatments: 100.50 | | |
| | | 5 August–7 August | 4.38 | 4.44 | 4.40 |
| Filling-Ripening stage | 5 August–8 September | | Covers all three treatments: 34.80 | | |
| | Total Irrigation from Filling to Ripening Stage | | 139.68 | 139.74 | 139.70 |
| | Irrigation Deviation | | −0.32 | −0.26 | −0.30 |
| Total Water Supply (Irrigation Plus Precipitation) in Whole Growing Season (No Decimals) | | | 500.00 | 500.00 | 500.00 |
| Total Precipitation in Whole Growing Season | | | Covers all three treatments: 261.60 | | |
| Total Irrigation Only in Whole Growing Season (No Decimals) | | | 239.00 | 239.00 | 239.00 |

Note: N represents no irrigation; + represents irrigation increment and needs to be subtracted from the next irrigation; − represents irrigation loss, and needs to be added at the next irrigation. Date between precipitation and irrigation is continuous because the precipitation during the days of irrigation is automatically counted as the amount of precipitation in the interval between that irrigation and the next irrigation.

## 2.4. Field Management

The field was tilled approximately 1 week before sowing. At the time of tilling, a basal dose of fertilizer was evenly and equably distributed in the topsoil at a rate of 375 kg ha$^{-1}$ (1.35 kg per plot) of diammonium phosphate (N-P$_2$O$_5$-K$_2$O, 18-46-0) based on the N and P requirements; the fertilizer was applied in spade slits to avoid loss over the soil surface and sprinkled near the maize roots to ensure full absorption by the crops. No pesticides and insecticides were used during the whole growth period of maize to prevent the test results from being affected.

## 2.5. Climate Data

Slight fluctuations in temperature occurred among the different growth stages. The mean temperature was 24.60 °C (Figure 3). Relative humidity was significantly higher in July and August, reaching a maximum of 100%, compared with other months (Figure 3). It was found that, from the calculation of the food and agriculture organization (FAO) Penman–Monteith Equation (2), reference crop evapotranspiration (ET$_o$) and precipitation were 501 and 261 mm, respectively (Figure 4). The average ET$_o$ was 3.61 mm d$^{-1}$, with remarkable seasonal variation; ET$_o$ increased from April to July and then declined significantly after August, along with the decrease in solar radiation intensity and temperature. In August, lower ET$_o$ was positively related to lower temperature and higher relative humidity.

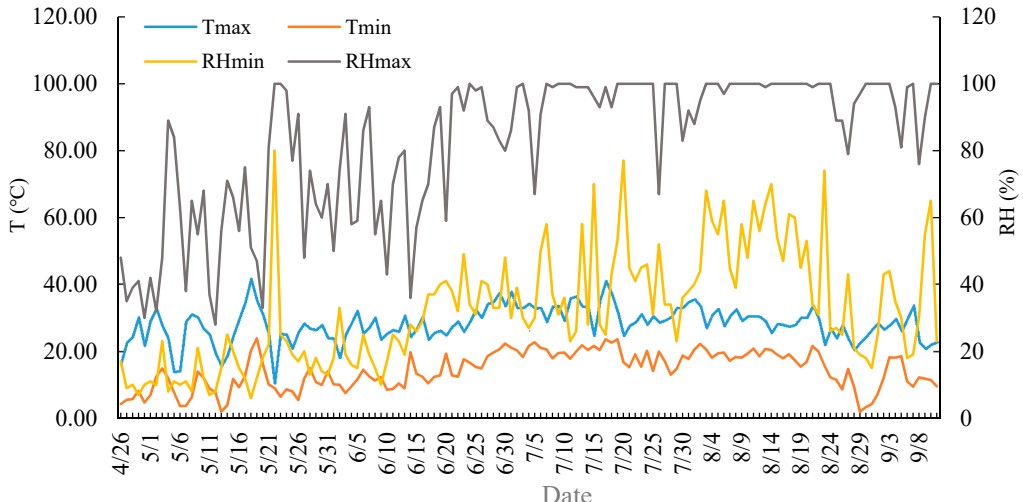

**Figure 3.** Variation of the temperature and relative humidity of the experimental site during the growth period of maize. Note: $T_{max}$ represents the maximum of temperature; $T_{min}$ represents the minimum of temperature; $RH_{max}$ represents the maximum of relative humidity; $RH_{min}$ represents the minimum of relative humidity.

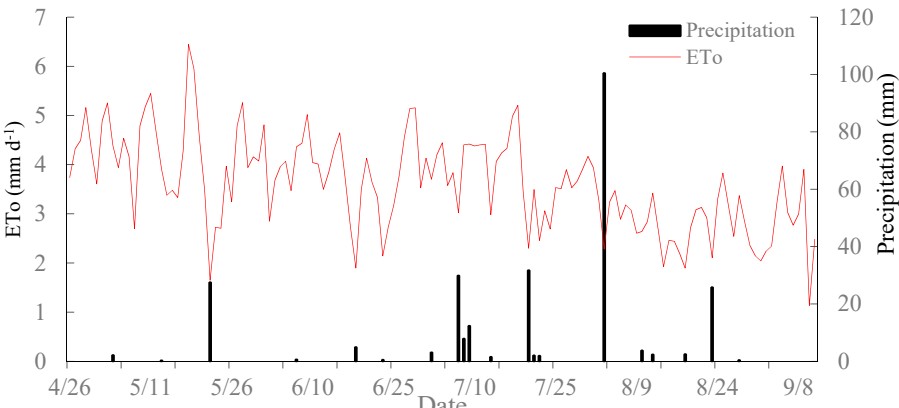

**Figure 4.** Variation of the reference crop evapotranspiration and precipitation during the growth period of maize. Note: $ET_o$ represents reference crop evapotranspiration, mm. $ET_o$ was calculated by means of the FAO Penman–Monteith Equation (2) for the entire growth season of maize.

### 2.6. Measurement of Indicators

Soil water content of 0–20 cm, 20–40 cm, 40–60 cm, 60–80 cm, and 80–100 cm in each growth stage was determined by the gravimetric method.

Soil temperature in the soil layer of 0–20 cm was measured by a thermometer during each growth period of maize.

Maize of each plot was harvested at the ripening stage, and 6 ears that were growing well were selected randomly in each plot. Drying the grain to constant weight at 85 °C, weighing by an electric balance for grain yield, the grain yields were then converted to a standard grain water content of 15.50% wet basis [27].

$ET_c$ was calculated daily during the growing season by the soil water balance Equation (1) [28]:

$$ET_c = I + P + C_r - D_w - R_f \pm \Delta s \ldots \tag{1}$$

where $ET_c$ was the total amount of actual evapotranspiration for the entire season (mm), $I$ was the amount of irrigation water applied (mm), $P$ was the precipitation (mm), $C_r$ was the capillary rise (mm), $D_w$ was the amount of drainage water (mm), $R_f$ was the amount of runoff (mm), and $\Delta s$ was the

change in the soil moisture content (mm). The soil moisture content measurement was used by the conventional oven-dry method in soil layers (0–20 cm, 20–40 cm, 40–60 cm, 60–80 cm, and 80–100 cm). No runoff was observed during the trials. Capillary rise was considered as negligible due to the deep water table level. Drainage water included precipitation under the effective rooting depth, according to the soil water content measurements in the soil layer at the effective rooting depth, was determined.

$ET_0$ was calculated per day during the growing season by using the FAO Penman–Monteith equation. The FAO Penman–Monteith equation is given by (2):

$$ET_0 = \frac{0.408\Delta(R_n - G) + \gamma\frac{900}{T+273}\mu_2(e_s - e_a)}{\Delta + \gamma(1 + 0.34\mu_2)} \tag{2}$$

where $ET_0$ was the reference evapotranspiration (mm day$^{-1}$), $R_n$ was net radiation at the crop surface (MJ m$^{-2}$ day$^{-1}$), $G$ was soil heat flux density (MJ m$^{-2}$ day$^{-1}$), $T$ was mean daily air temperature at 2 m height (°C), $\mu_2$ was wind speed at 2 m height (m s$^{-1}$), $e_s$ was the saturation vapor pressure (kPa), $e_a$ was the actual vapor pressure (kPa), $e_s$ - $e_a$ was the saturation vapor pressure deficit (kPa), $\gamma$ was the slope of the saturation vapor pressure curve (kPa °C$^{-1}$), and $\Delta$ was the psychrometric constant (kPa °C$^{-1}$). Meteorological parameters needed to calculate $ET_0$ were derived from a local meteorological station.

Water use efficiency (kg m$^{-3}$) was calculated as (3) [29]:

$$WUE = Y/ET_c \ldots \tag{3}$$

where $WUE$ was the water use efficiency (kg m$^{-3}$), $Y$ was the grain yield (kg hm$^{-2}$), $ET_c$ was the total amount of actual evapotranspiration for the entire season (mm).

Net economic profit (yuan hm$^{-2}$) was calculated as (4):

$$\text{Net profit} = \text{total revenue} - \text{total cost} \tag{4}$$

where, total revenue = grain yields × average local price. The average local price for maize was 1.70 yuan kg$^{-1}$. The total cost included the cost of seed, fertilizers, sows, micro-spray irrigation belts, water pipes, maize straw, and organic fertilizers during the trial.

Light and temperature production potential (kg hm$^{-2}$) was calculated by the photosynthetic production potential multiplied by the revised function of temperature effect, and the expression is given by (5):

$$Y_c = Y_p \cdot f(T) \ldots \tag{5}$$

$Y_c$ was the light and temperature production potential, kg hm$^{-2}$;
$Y_p$ was the photosynthetic production potential, kg hm$^{-2}$;
$f(T)$ was the revised function of the temperature effect.
Water saving potential per unit yield (m$^3$ kg$^{-1}$) was calculated as (6):

$$P_y = \frac{1}{WUE_a} - \frac{1}{WUE_t} \ldots \tag{6}$$

$P_y$ was water saving potential per unit yield, m$^3$ kg$^{-1}$;
$WUE_a$ was actual crop water productivity, kg m$^{-3}$;
$WUE_t$ was the theoretical crop water productivity, kg m$^{-3}$.
Water saving potential per unit area (m$^3$ hm$^{-2}$), was calculated as (7):

$$S_p = P_p \times W \ldots \tag{7}$$

$S_p$ was the water saving potential per unit area, m$^3$ hm$^{-2}$;
$P_p$ was the proportion of crop water saving potential, %;
$W$ was the irrigation quota, m$^3$ hm$^{-2}$.

### 2.7. Data Collations and Statistical Methods

Effects of various surface mulching patterns on soil water content, soil temperature, yield, water consumption, water saving potential, and net profit of maize were plotted using Origin 8.0. Variance, correlation and stepwise regression analyses were performed using SPSS 20.0, while significant differences were detected using the least significant difference (LSD) test. The cost and net profit of various surface mulching patterns were compared using the quota comparison method. Tables were created in Excel 2010.

## 3. Results

### 3.1. Changes in the Soil Temperature and the Soil Water Content

Under micro-spray irrigation, variation in soil temperature and soil water content during the entire growth period of maize was not affected by the different surface mulching patterns. Soil temperature increased from the seedling stage to the heading stage, which remained relatively stable until the filling stage, and then declined. In each treatment, the soil water content was significantly higher at heading and maturity than at other growth stages (Figure 5).

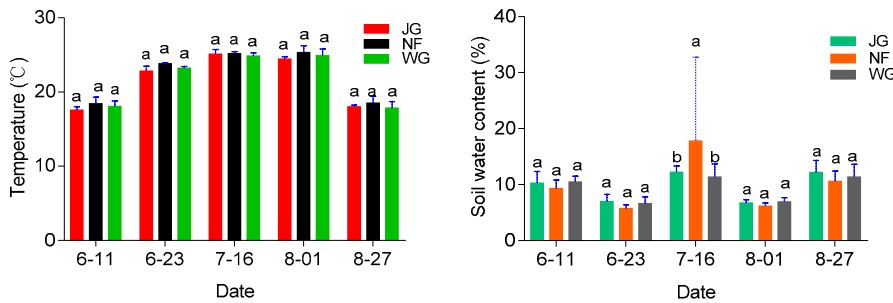

**Figure 5.** Soil temperature (soil layer of 0–20 cm) and soil water content (soil layer of 0–100 cm) of maize at different growth stages. Note: JG represents straw mulching; NF represents organic fertilizer mulching; WG represents no mulching; data represents mean ± standard error ($n = 17$). Bars labeled with different letters (lowercase) differed significantly among the treatments ($p < 0.05$).

The temperature of the 0–20 cm soil layer was the highest from the seedling stage to the heading stage, and the rate of increase of soil temperature reached 42.99%, 36.71%, and 37.55% in the JG, NF, and WG treatments, respectively (Figure 5). Then, the soil temperature decreased from the heading to the ripening stage by 28.29%, 26.51%, and 28.24% in the JG, NF, and WG treatments, respectively. Overall, during the entire growth period of maize, the mean soil temperature (0–20 cm layer) in different treatments was in the order: NF > WG > JG.

Soil water content at each growth stage was lower in JG and WG treatments than in the NF treatment (Figure 5); thus, the soil water content was directly related to the variation in soil temperature in different mulching treatments. However, there was no significant difference between JG, and WG, respectively, for the mean soil water content. In all three treatments, the soil water content was the highest at the heading stage, which was directly responsible for the higher irrigation proportion (35% of the total water supply). Compared with the seedling stage, the heading stage showed an increase in soil water content of 18.70%, 91.05%, and 8.33% in the JG, NF, and WG treatments, respectively.

### 3.2. Water Saving Potential (Per Unit Yield and Per Unit Area)

Values of $P_y$ and $S_p$ of maize in the JG treatment were significantly lower than those in the NF and WG treatments. Compared with the JG treatment, values of $P_y$ and $S_p$ were significantly higher by 31.73% and 34.51%, respectively, in the NF treatment and by 20.21% and 20.08%, respectively in the WG treatment ($p < 0.05$); however, no significant differences were detected in $P_y$ and $S_p$ between the

NF and WG treatments. Thus, under micro-spray irrigation, $P_y$ and $S_p$ of maize in various treatments were in the order: NF and WG >JG (Figure 6).

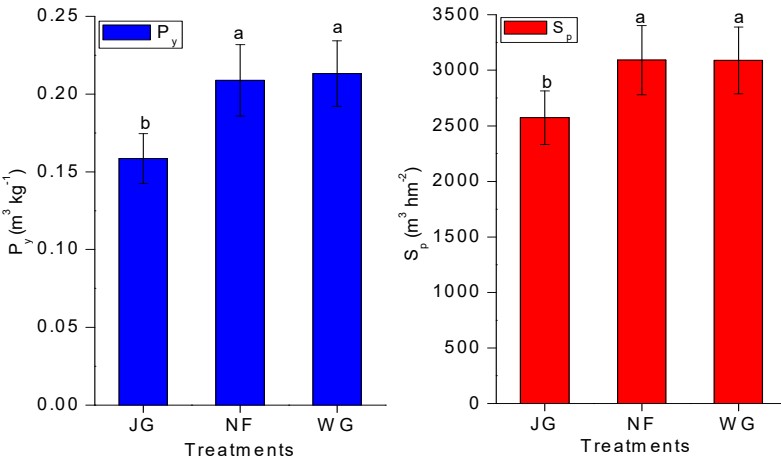

**Figure 6.** Water saving potential per unit yield and per unit area of maize under various surface mulching patterns. Note: JG represents straw mulching; NF represents organic fertilizer mulching; WG represents no mulching; $P_y$ represents water saving potential per unit yield, $m^3$ $kg^{-1}$; $S_p$ represents water saving potential per unit area, $m^3$ $hm^{-2}$. Bars labeled with different letters (lowercase) differed significantly among the treatments ($p < 0.05$).

The *P* regression model affecting the water-saving potential per unit yield and per unit area was 0.00, $R^2$ was up to 0.92, 1.00, respectively, with high fitting degree (Table 2). This elucidated that under various surface mulching patterns, the index that affected $P_y$ was yield ($R^2 = 0.92$), and indexes that affected $S_p$ were light and temperature production potential ($Y_c$), yield and water consumption ($ET_c$).

**Table 2.** Stepwise regression coefficient of indexes affecting water saving potential of maize.

| Indicator | Model | Regression Coefficient | t | *P* | $R^2$ |
|---|---|---|---|---|---|
| $P_y$ | (Constant) | 0.696 | 13.358 | 0.000 | 0.920 |
|  | Y | $-3.320 \times 10^{-5}$ | −9.648 | 0.000 |  |
| $S_p$ | (Constant) | 299.478 | 102.482 | 0.000 |  |
|  | $Y_c$ | 0.057 | 2029.233 | 0.000 | 0.988 |
|  | Y | −0.151 | −1866.520 | 0.000 |  |
|  | $ET_c$ | −0.237 | 3.857 | 0.012 |  |

Notes: $P_y$ represents water saving potential per unit yield, $m^3$ $kg^{-1}$; $S_p$ represents water saving potential per unit area, $m^3$ $hm^{-2}$; $Y_c$ represents light and temperature production potential of crop, kg $hm^{-2}$; Y represents grain yield, kg $hm^{-2}$; $ET_c$ represents water consumption, mm. t represents significance test values of regression parameters. *P* represents significant value.

### 3.3. Light and Temperature Production Potential, Yield and Water Consumption

The JG treatment showed the lowest $ET_c$ and the highest yield and WUE (Figure 7); however, no significant differences were detected in these parameters between the NF and WG treatments. The value of $ET_c$ was essentially the same in JG, NF, and WG treatments. Values of $Y_c$ in the NF and WG treatments were 16.18% and 13.32% higher than those in the JG treatment, respectively ($p < 0.05$). However, maize yield in the JG treatment was 9.53% and 12.10% higher than that in the NF and WG treatments ($p < 0.05$). In the current study, $ET_c$ was approximately 500 mm in the JG, NF, and WG treatments during the entire growth season; no significant differences were detected among them, probably because of the thickness of maize straw in the JG treatment and organic fertilizer in the NF treatment, which requires further investigation. Moreover, this difference was closely related to differences in the region, maize variety, and irrigation approach.

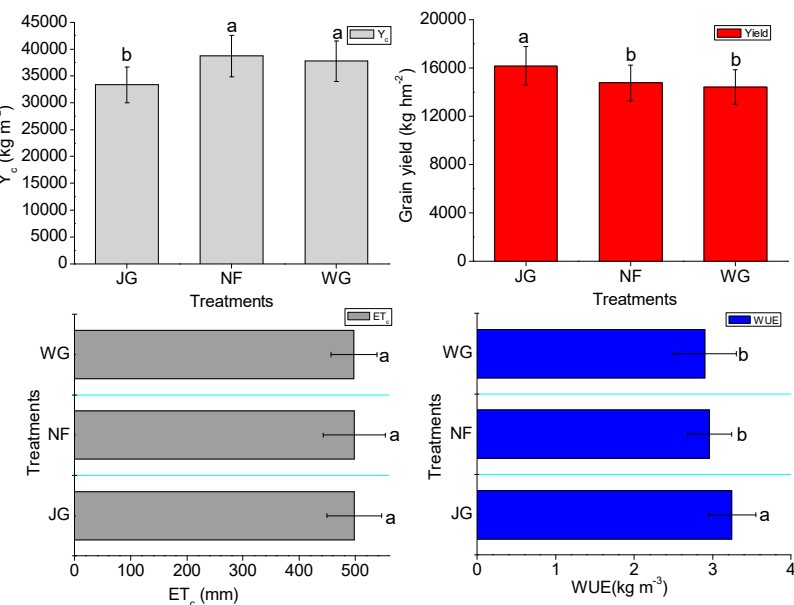

**Figure 7.** Light and temperature production potential, yield, water use efficiency and water consumption of maize under various mulching patterns. Note: JG represents straw mulching; NF represents organic fertilizer mulching; WG represents no mulching. $Y_c$ represents light and temperature production potential; $ET_c$ represents water consumption of maize; WUE represents water use efficiency. Bars labeled with different letters (lowercase) differed significantly among the treatments ($p < 0.05$).

In the JG, NF, and WG treatments, correlation coefficients between water saving potential ($P_y$ and $S_p$) of maize and indexes affecting water saving potential were greater than 0.80 (Table 3), indicating a high correlation. Values of $P_y$ and $S_p$ changed significantly with yield ($R > 0.90$).

**Table 3.** Correlations between indexes affecting water saving potential.

| Index | Treatment | Y (kg hm$^{-2}$) | Index | $Y_c$ (kg m$^{-2}$) | Y (kg hm$^{-2}$) | $ET_c$ (mm) |
|---|---|---|---|---|---|---|
| $P_y$ (m$^3$ kg$^{-1}$) | JG | 0.951 * | $S_p$ (m$^3$ hm$^{-2}$) | −0.982 ** | 0.905 * | 0.887 * |
| | NF | −0.999 ** | | −0.982 ** | −0.982 ** | −0.997 ** |
| | WG | 0.982 ** | | −0.817 * | 0.982 ** | 1.000 ** |

Notes: * represents the significant difference at the level of 0.05 (bilateral); ** represents the significant difference at the level of 0.01 (bilateral).

### 3.4. Economic Profits

Analysis of the economic profits of various treatments showed that the WG treatment with micro-spray irrigation was the least expensive (9575.33 yuan hm$^{-2}$; Figure 8) because this treatment had no associated cost of surface mulching material. However, the WG treatment showed no significant difference compared with the JG and NF treatments because of the higher labor cost associated with no mulching in the WG treatment. Net profit was the highest in the JG treatment, which was significantly higher by 18.26% and 17.71% than those in the NF and WG treatments, respectively ($p < 0.05$). The profit:cost ratio was 1.78 in the JG treatment, which was 22.14% and 13.93% higher than that in the NF and WG treatments, respectively. According to the regression fitting analysis, the net profit of maize showed a significantly negative linear correlation with $P_y$ and $S_p$, with the coefficient of determination ($R^2$) of 0.898 and 0.989, respectively (Figure 9). The higher the water saving potential, the smaller the net economic profit of maize; this explained why the net economic profit of maize in the JG treatment was higher than that in the NF and WG treatments.

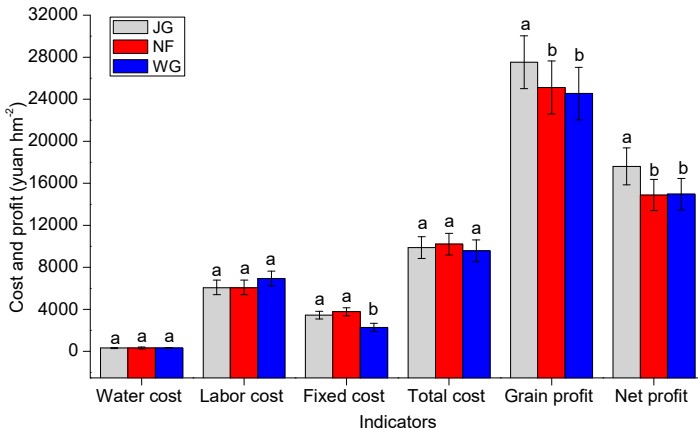

**Figure 8.** Cost and profit of maize under various mulching planting patterns. Note: JG represents straw mulching; NF represents organic fertilizer mulching; WG represents no mulching. The average local price for maize was 1.70 yuan kg$^{-1}$. The total cost included the cost of seed, fertilizers, sows, micro-spray irrigation belts, water pipes, maize straw, organic fertilizers during the trial. Labor costs included the layout of the sample plot, weeding, sampling, irrigation, spreading fertilizer, and determination of sample. Bars labeled with different letters (lowercase) differed remarkably among different indexes ($p$ <0.05).

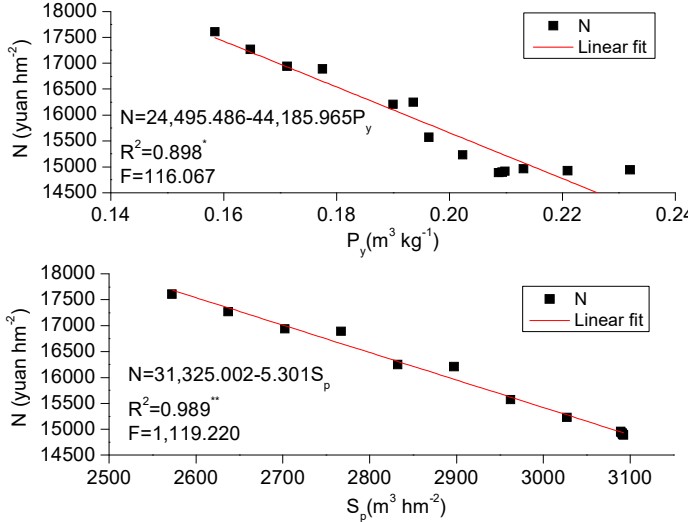

**Figure 9.** Relationships between water saving potential and net economic profit of maize. Note: N represents net economic profit, yuan hm$^{-2}$; $P_y$ represents water saving potential per yield, m$^3$ kg$^{-1}$; $S_p$ represents water saving potential per unit area, m$^3$ hm$^{-2}$. * represents regression effect was significant; ** represents regression effect was remarkably significant.

## 4. Discussion

### 4.1. Water Saving Potential

Currently, climate change is one of the major concerns of the environmental problem facing mankind, and agriculture is highly sensitive to climate change [30]. Temporal and spatial distribution patterns of climate resources, such as water and heat, are directly affected by meteorological factors, including solar radiation, temperature, and precipitation. Fluctuations in temperature and precipitation during the growing season affect crop productivity, ultimately affecting regional agricultural production [31]. Therefore, spatiotemporal distribution characteristics of the crop climate production potential represent the basis of comprehensive food production potential and provide a crucial theoretical basis for agricultural productivity planning and agricultural structural

adjustment [32]. Water saving potential is a vital evaluation index for adjusting agricultural structure. In this study, soil water content at each growth stage was lower in JG and WG treatments than in the NF treatment (Figure 5). However, there was no significant difference between JG, and WG, respectively, for the mean soil water content. This resulted mainly from the type and rate of the irrigation and the mulches. Because the soil moisture strongly depended on the precipitation and irrigation pattern (see irrigation scheme), the precipitation, irrigation pattern, and irrigation amount applying to all three treatments were the same. In addition, Zhao et al. [33] found that, compared with deep tillage with no mulch, mean soil water content of sunflowers was only higher by 5.75%, 2.50%, respectively, in 2011 and 2013, when straw mulch was used at a rate of 12,000 kg hm$^{-2}$, which was significantly higher than that of our study (4000 kg hm$^{-2}$). Teame et al. [34] indicated that, by exploring the efficacy of organic mulching, sorghum straw mulching and rice straw mulching with a rate of 10,000 kg hm$^{-2}$ increased mean water soil content of sesame by 33.29%, 42.05%, respectively, compared to no mulch. The efficiency of increasing soil water was more significant than that in Zhao et al. [33], which would be due to the difference in the crops and the mulches. Therefore, it was suggested that yield, $P_y$, and $S_p$ were not affected significantly by the water soil content, based on the results of the stepwise regression analysis (Table 2) and the insignificant difference among the three treatments for water soil content (Figure 5). The water saving potential was mainly affected by $Y_c$, grain yield and $ET_c$, and showed a positive correlation with $Y_c$ and negative correlation with grain yield and $ET_c$. Therefore, both $P_y$ and $S_p$ were the lowest in the JG treatment; however, the WUE and water saving capacity of the JG treatment were significantly higher than that of the NF and WG treatments.

Photosynthetic production potential represents the maximum crop yield achieved only under light conditions. $Y_c$ was the maximum yield of crop subjected to both light and temperature constraints [35]. Regions that experienced a rapid decrease in the climate production potential of maize over the last 30 years show a dramatic increase in $Y_c$, resulting from a dry climate [32]. Radiation and temperature are the most critical factors affecting $Y_c$; these factors decreased by −12.70% and −6.10%, respectively, when solar radiation of maize decreased by 10% or temperature increased by 1 °C during the growing season [35]. Under micro-spray irrigation, maize yield was the highest in the JG treatment, in which $Y_c$ was considerably lower than that in the NF and WG treatments, although no significant differences were detected in $ET_c$ among the three treatments. The photosynthetic production potential of the yield has been the main focus of research in northeast China [36]. The decrease in solar radiation is primarily responsible for the decrease in the photosynthetic production potential of maize. Spatial variation characteristics of the photosynthetic production potential of maize were similar to those of the surface solar radiation, both of which showed a decreasing trend from the southwest to the northeast [37]. Licker et al. systematically analyzed global maize yields and concluded that the maize yield would increase by 50%, if 95% of the maize cultivation areas worldwide met the climate potential. However, the difference between climate production potential and actual production was approximately 20% because of improvements in traditional farming patterns, economic costs, and technological measures, and the yield potential of maize was mainly affected by unreasonable farmland management and low technical levels. Therefore, conclusions explored in our study on the water saving potential of maize using different planting patterns and micro-spray irrigation are extremely conducive to the rational planning of the layout of planting areas, effectively improving maize yield and ensuring food security.

### 4.2. Yield and Water Consumption

Mulching has been adopted in numerous parts of the world as an approach to increase crop productivity [38]. Maize yield in the JG treatment was 9.53% and 12.10% higher than that in the NF and WG treatments ($p$ <0.05), and was significantly higher than that reported by Sharma et al. [39] and similar to that reported by many scholars. Li and colleagues [40] showed that plastic mulching dramatically increases crop yield. Yin et al. [41] found that compared to conventional tillage without straw residue, integrating no tillage with two-year plastic and straw mulching improved grain yields by 13.8%, reduced soil evaporation by 9.0%, and reduced soil evaporation by 9.0%. However,

in another study, straw mulching did not significantly affect yield under limiting soil water content [42]. In addition, $ET_c$ was approximately 500 mm in the JG, NF, and WG treatments during the entire growth season; no significant differences were detected among them, probably because of the thickness of maize straw in the JG treatment and organic fertilizer in the NF treatment, which requires further investigation. Our results of $ET_c$ were in contrast to those of previous studies. For example, Yin et al. [43] reported that plastic film together with straw mulching decreased total evapotranspiration by an average of 4.60% ($p$ <0.05) compared with no mulching; Brar et al. [44] suggested that straw mulching resulted in 19.00% higher yield compared with no mulching, resulting in 36.20 mm higher transpiration and 44.20 mm lower soil evaporation; Sun et al. [45] confirmed that crop water consumption was reduced by 32 mm under straw mulching compared with no mulching, with no significant differences in WUE. Zhou et al. [46] indicated that compared with no mulching, straw mulching increased maize yield by 10.6% and 12.5% under a drip irrigation system in 2016 and 2017, respectively, and achieved 6.1% lower water consumption. The contradiction was mostly responsible for the enhanced maize straw mulching amount in their study, which was significantly higher than that of our study. Moreover, this difference was closely related to differences in the region, maize variety, and irrigation approach.

### 4.3. Economic Profits

Economic profits include the rate of input application and the rate of consumptive use in irrigation and fertilizer [47]. Net profit was the highest in the JG treatment, which was significantly higher by 18.26% and 17.71% than those in the NF and WG treatments, respectively ($p$ <0.05). Similar net profit has been reported in rice using straw mulching in water saving production systems [48]. In this study, the profit: cost ratio was 1.78 in the JG treatment, which was 22.14% and 13.93% higher than that in the NF and WG treatments, respectively. This was consistent with the results of Sharma et al. [39]; the authors showed that the profit: cost ratio was the highest (0.62) with straw mulching, although this ratio was markedly lower than that obtained in our study. In other studies, drip irrigation resulted in net economic profits of 4359.58–6240.19 yuan $hm^{-2}$, which were higher than those obtained by furrow irrigation [44]. Small amounts of maize cob biochar would also attain higher net profit through increased yields [49]. Sweet maize and green beans grown in rotation resulted in a greater increase in net profits compared with potato monoculture [50]. Therefore, straw mulching with micro-spray irrigation elevated furthest economic profits of maize, compared to organic manure mulching and no mulching.

## 5. Conclusions

Under micro-spray irrigation, maize yield and WUE were the highest in the JG treatment with 16,178.40 kg $hm^{-2}$, 3.25 kg $m^{-3}$, respectively, in which $Y_c$ was significantly lower than that in the NF and WG treatments. The three treatments showed no significant differences in $ET_c$. The water saving potential (including $P_y$ and $S_p$) of maize was positively affected by $Y_c$ and negatively affected by grain yield and WUE. Therefore, values of $P_y$ and $S_p$ were the lowest in the JG treatment, were just 0.16 $m^3$ $kg^{-1}$ and 2572.31 $m^3$ $hm^{-2}$, respectively, but no significant differences were found, compared to NF and WG treatments. The net economic profit of maize was negatively correlated with the water saving potential in all treatments, which was primarily responsible for the negative relationship between water saving potential and yield, WUE, respectively. So, the maximum of net economic profit appeared in the JG treatment, was up to 17,610.09 yuan $hm^{-2}$, and was higher than that in the NF and WG treatments ($p$ <0.05).

The yield, WUE, and water saving capacity of the JG treatment were significantly higher than that of the NF and WG treatments. This suggests that straw mulching with micro-spray irrigation should be applied in local appropriate farmland. Given the lower WUE and higher water saving potential of the NF and WG treatments, it is important to explore these planting patterns further.

**Author Contributions:** Z.H. designed, performed the experiments, and wrote the original draft; X.S. analyzed the data; T.Z. reviewed and edited the draft. All authors read the final manuscript and approved the submission. All authors have read and agreed to the published version of the manuscript.

**Funding:** This research was funded by the National Natural Science Foundation of China (Grant No. 41371053, 30972422, 51669034, 51809224), National key research and development project of China (Grant No. 2017YFC0506706, 2017YFC0504704) and Science research launch project of PhD (205040305).

**Acknowledgments:** We are grateful to all the members of Naiman Desertification Research Station, Chinese Academy of Sciences, for their help in field work. We are also grateful to other anonymous reviewers for their valuable comments on the manuscript.

**Conflicts of Interest:** The authors declare no competing interests.

## Abbreviations

| | |
|---|---|
| $ET_o$ | REFERENCE CROP EVAPOTRANSPIRATION |
| $ET_c$ | WATER CONSUMPTION |
| FAO | FOOD AND AGRICULTURE ORGANIZATION |
| JG | STRAW MULCHING |
| NF | ORGANIC MANURE MULCHING |
| $P_y$ | WATER SAVING POTENTIAL PER UNIT YIELD |
| $S_p$ | WATER SAVING POTENTIAL PER UNIT AREA |
| WG | NO MULCHING |
| WUE | WATER USE EFFICIENCY |
| $Y_c$ | LIGHT AND TEMPERATURE PRODUCTION POTENTIAL |

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
