# Peer review of "Study on Water Saving Potential and Net Profit of Zea mays L.: The Role of Surface Mulching with Micro-Spray Irrigation"

_applsci, doi:10.3390/app10010402_

Round 1

Reviewer 1 Report

The authors have done a good job revising the manuscript.  I have made some more comments.  In some places, I have highlighted words with yellow which they need to pay more attention during the revision process. Other comments are directly inserted in the manuscript. Please see the attached file.

Reviewer 2 Report

The manuscript needs to be better presented as the message delivered to the reader is not clear. Methods need to be better described. Some concepts need to be better explained while others find their natural place in the Introduction. In my opinion the manuscript warrants publication after major revision as next specified:

LINE 18. I deduce that the total amount of water (500 mm) was given by both irrigation and precipitation. Since precipitation is a casual event, is it correct that the quantity provided by irrigation is to reach the total due of 500 mm? if so, please add a brief discussion in the abstract to make it clearer and extend it at LINE 107.

LINE 51. “Water-saving potential refers to the amount of water that has not been saved”: is it a definition or a common shared opinion? If so, try to give a reference. The concept is recalled later on, e.g. at LINE 62

FIGURE 1. Improve resolution.

The cited literature is comprehensive but somewhere not up-to-date; I would suggest to take also into account the following research papers:

Hengsdijk H., Guanghuo W., Van den Berg M.M., Jiangdi W., Wolf J., Changhe L., Roetter R.P., van Keulen H. (2007) Poverty and biodiversity trade-offs in rural development: a case study for Pujiang County, China, Agric. Syst. 94, 851–861.

Huang Q., Rozelle S., Lohmar B., Huang J., Wang J. (2006) Irrigation, agricultural performance and poverty reduction in China, Food Policy 31, 30–52

Another work of Yin: Yin, W., Fan, Z., Hu, F., (...), Chai, Q., Coulter, J.A. (2019). Innovation in alternate mulch with straw and plastic management bolsters yield and water use efficiency in wheat-maize intercropping in arid conditions. Scientific Reports 9(1),6364

Zhou, B., Ma, D., Sun, X., (...), Ma, W., Zhao, M. (2019). Straw mulching under a drip irrigation system improves maize grain yield and water use efficiency. Crop Science. 59(6), pp. 2806-2819

Wang, C., Wang, J., Xu, D., (...), Sui, J., Mo, Y. (2019). Water consumption patterns and crop coefficient models for drip-irrigated maize (Zea mays L.) with plastic mulching in northeastern China. Transactions of the ASABE. 62(3), pp. 571-584

LINE 91. “Each plot contained four rows, with six maize plants per row, at a plant-to-plant spacing of 24 cm and row-to-row spacing of 40 cm (Figure 2)”. Could you justify these spacing given with the order of centimetres. Please explain why 1m is judged large enough to avoid edge effects.

LINE 107. “Maize plants were supplied with 107 mm of water, including irrigation and precipitation,” See Line 18.

LINE 109. “According to the water requirements”. If possible, give references.

LINE 114. “with a water-supply pressure of 0.2 MPa” which kind of information is given here? In my opinion this is unnecessary.

LINE 114. “Maize was irrigated on days with no or low wind (<1.5 m s-1) to achieve uniform irrigation.” In other terms, the 500 mm of water, split in the percentages 15%, 35%, 22% and 28% are not given continuously in time depending if the generic day is windy or not. A discussion about possible side effects is encouraged.

LINE 132. I guess <ET0>=3.61 mm d^-1 is calculated by means of Equation 2. If so, provide this explanation.

LINE 145. “Soil water content of 0-20cm, 20-40cm, 40-60cm, 60-80cm and 80-100cm”. Soil moisture strongly depends on precipitation and irrigation pattern (see Line 114). A discussion is welcomed.

LINE 154. Equation 2.1 should be renamed as Equation 1.

LINE 252. “Soil water content of 0-20cm, 20-40cm, 40-60cm, 60-80cm and 80-100cm” I would move this sentence back in the Introduction. In general, I see here some discussion (and related references) that can be conveniently moved in the Introduction as 3.3 is a Result paragraph.

LINE 320. I see that most of the contents here are not related to the discussion of the obtained results as it should be instead. This part should be completely rethought.  

LINE 358. Conclusions need to be more specific and extended.

Round 2

Reviewer 2 Report

Dear Authors, the manuscript has been significantly improved. I acknowledge the efforts made by you.